# Toxic Elements Behavior during Plasma Treatment for Waste Collected from Power Plants

Ibrahim AlShunaifi [1], Imed Ghiloufi [2,*], Abdullah Albeladi [1] and Ahmed Alharbi [1]

[1] King Abdulaziz City for Science and Technology (KACST), National Center for Combustion & Plasma Technology, Riyadh 11442, Saudi Arabia; ialshunaifi@kacst.edu.sa (I.A.); aalbeladi@kacst.edu.sa (A.A.); aalharbi@kacst.edu.sa (A.A.)

[2] College of Sciences, Imam Mohammad Ibn Saud Islamic University (IMSIU), Riyadh 11623, Saudi Arabia

[*] Correspondence: ioghiloufi@imamu.edu.sa

**Abstract:** The subject of this work is the treatment of solid waste collected from power plants using thermal plasma technology. Inductively coupled plasma (ICP), X-ray fluorescence (XRF), and energy-dispersive X-ray spectroscopy (EDX) were used to characterize the waste before and after the treatment. The results show that waste is formed essentially from carbon, but it also contains sulfur and toxic elements like lead, cadmium, zinc, and arsenic. For this reason, a plasma reactor was used to separate carbon from the heavy metals by a pyrolysis/combustion plasma system. After the plasma treatment, the mass of the waste was reduced by more than 85% and the metals were collected in the filter bag. A computer code was used to study the toxic element volatility during the treatment. With this code, the effects of plasma temperature, confinement matrix, and the composition of the carrier gas on the volatility of lead and arsenic were determined. The code results show that arsenic remains in the liquid phase for temperatures less than 2000 K, whereas for temperatures beyond 2100 K, arsenic becomes very volatile. For lead, any increase in temperature increases its vaporized quantity and its vaporization speed. The addition of oxygen in the carrier gas leads to the heavy metal incorporation in the confinement matrix. The increase of the quantity of Ba in the containment matrix strengthens the confinement of as in the matrix.

**Keywords:** simulation; toxic elements; thermal plasma; volatility; waste





## 1. Introduction

The waste that is generated from power plants that use oil as a source of fuel contains generally a high percentage of carbon, alkaline-earth metals, and metals like Co, Fe, and Mn [1]. V, Pb, Ni, Cd, and Cr are also present in this fly ash, and for this reason it is classed as toxic waste [2–5]. Numerous methods for fly ash processing exist in order to reduce the volume and impact of the waste [6]. Fly ash processing can usually be divided into a number of categories, including storage, chemical processing, and thermal processing, like incineration, pyrolysis and gasification [7].

Conventional incineration or combustion of waste has long since held a negative public perception for being environmentally unfriendly. In comparison to incineration, newer technologies like pyrolysis and gasification are preferred [8]. Plasma treatment differs from conventional thermal conversion methods because of the addition of high density energy into the system by means of the plasma torch or plasma arc [9]. During conventional processes, all the energy used for conversion is taken from the feedstock itself. Plasma energy addition thus allows for increased operating temperatures, higher conversion rates and better control of the overall process. For these reasons, plasma technology becomes a promising technique for the treatment of most toxic wastes, such as contaminated hospital wastes, radioactive wastes, and fly ash [10–12]. One of the plasma processing inconveniences is the volatility of toxic elements during the treatment. For this

reason, the volatility of heavy metals needs be controlled during the plasma treatment of waste.

During the treatment of wastes by thermal plasma, the concentrations of toxic elements, such as heavy metals and radioelements, in the gas phase were measured using optical emission spectroscopy method [13]. Furthermore, many computer codes were developed to simulate the volatility of toxic elements present in the waste during their plasma treatment. For example, a computer code was developed to simulate the volatility of heavy metals during fly ash treatment using a plasma arc [14]. This same code was used to study the effect of different parameters, such as bath surface temperature, plasma current, and partial pressure of oxygen, on the volatility of radioelements during thermal plasma treatment of radioactive waste [10,15,16].

This study has two objectives; the first one is to treat the waste collected from power plants in Saudi Arabia using our new plasma furnace. This treatment consists of the separation of carbon from metals presents in the waste. The second objective of this work is to control the volatility of toxic elements, such as lead and arsenic, during the treatment using a computer code.

## 2. Materials and Methods

### 2.1. Experimental Setup

The system used to treat the waste is shown in Figure 1. This system contains a plasma furnace that supports a plasma torch, exchanger, filter bag, and column wash. The non-transferred arc torch was placed inside a furnace and above a crucible that was filled with the waste. The fly ashes leaving the furnace (product) were collected in the filter bag, and they were re-introduced to the furnace to complete the combustion of the carbon. This process was repeated three times and in each cycle the composition of the product was analyzed by different techniques.

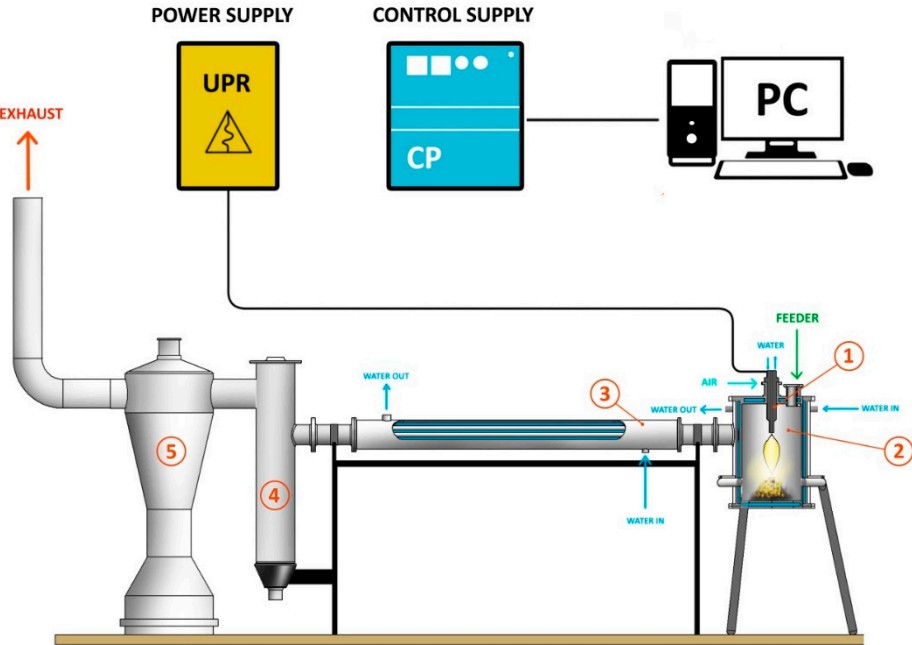

**Figure 1.** Plasma system for fly ash processing. 1. Plasma torch, 2. Furnace, 3. Heat exchanger, 4. Filter bag, 5. Scrubber.

Water under pressure was used to cool the torches and the reactor. In the experiment, the current was 100 A, the voltage was 220 V, and the carrier gas used for the air had a flow rate of 45 m$^3$/h.

## 2.2. Analysis Methods

To determine the waste chemical composition, X-ray fluorescence (XRF) was used. The concentrations of carbon and sulfur present in the waste were measured using energy-dispersive X-ray spectroscopy (EDX). Whereas, to measure the concentration of toxic elements, such as Pb, Cd, Zn, Cr, and As, inductively coupled plasma atomic emission spectroscopy (ICP-AES) was used

Inductively Coupled Plasma (ICP) is an analytical technique used for determining metal concentrations. ICP is a type of emission spectroscopy that uses double-induction plasma (argon, in our case) to produce excited atoms and ions that emit electromagnetic radiation at characteristic wavelengths of a particular element. The intensity of this emission indicates the concentration of the element within the sample.

The working principle of XRF is based on the excitation of the sample atoms by high-energy X-rays, followed by the emission of characteristic photons with a certain energy, well correlated to the atomic number Z of each element. The determination of the energy (or wavelength) of the emitted photon allows for qualitative analysis and the determination of the number of emitted characteristic photons allows for quantitative analysis. To perform XRF, a spectrometer is required that consists of an excitation source, sample and a detection system, which can be either wavelength-dispersive or energy-dispersive.

Energy Dispersive X-ray Analysis (EDX), referred to as EDS or EDAX, is an X-ray technique used to identify the elemental composition of materials. EDS makes use of the X-ray spectrum emitted by a solid sample bombarded with a focused beam of electrons to obtain a localized chemical analysis. Qualitative analysis involves the identification of the lines in the spectrum and is straightforward owing to the simplicity of X-ray spectra. Quantitative analysis entails measuring line intensities for each element in the sample and comparing this data to the known composition of these same elements.

## 2.3. Model Description

A computer code was developed to control the volatility of toxic elements during the thermal plasma treatment of the waste. This code allows the calculation of the mole numbers of each species present in the system. The species distribution was obtained by the total free energy minimization method, in which the equations of equilibrium were coupled to the partial pressures of oxygen and carbon [17,18]. The system is composed essentially of metallic oxides, carbides, and carbon oxides, and for this reason we must couple the equations of equilibrium with those of the mass transfer of oxygen and carbon. This coupling is assured by the fixation of partial pressures of oxygen and carbon. The matter conservation equation of oxygen was developed in our previous studies [19,20], whereas in this study, the equation of carbon mass transfer at the interface was introduced in the code and it has the following form:

$$\sum_{j=1}^{N} X_j (J_{M_j}^G) - \sum_{i=1}^{Ng} a_{ik} J_i^G - \frac{1}{A} \sum_{j=1}^{N} n_{M_j} \cdot \frac{dX_j}{dt} = 0 \tag{1}$$

where $n_{Mj}$ is the total mole number of metal $j$ in the liquid phase, $X_j$ represent the stoichiometric coefficient of a metal $j$, $a_{ik}$ is the carbon stoichiometric coefficient in species $i$, $J_{M_j}^G$ represents the molar flux density of metal $j$ in the gas state.

In each iteration, the code calculates the species mole number in two phases-liquid and gas. The amount of material found in the gas phase does not leave the furnace completely, but a certain quantity remains in equilibrium with the bath under the effects of electrolysis. Hence, the composition of gas was not obtained by a single liquid–gas equilibrium, but it is the result of the combination of diffusive transport, reactional balances, and the effects of electrolysis.

The following equation represents the flux density lost, in each iteration, for the gas species $i$:

$$J_i^L = J_i^D - J_i^R \tag{2}$$

where $J_i^D$ is the density of the diffusion flux of gas species $i$ and $J_i^R$ is the retained flux by the bath and it is given by [16]:

$$J_i^R = \frac{I}{A \cdot F \cdot v} n_i^0 \qquad (3)$$

where $F$ is the Faraday' constant, $I$ is the plasma current, A is the interface surface, $n_i^0$ is the initial mole number of a species $i$, and $v$ is the species valence number.

The equation of the diffusion flux density for species $i$ in the gas state is [19,20]:

$$J_i^G = -\frac{D_i}{RT} \left( \frac{P_i^x - P_i^w}{\delta_i} \right) + J \cdot P_i^w \qquad (4)$$

where $D_i$ and $\delta_i$ are the diffusion coefficient and the boundary layer thickness of species $i$ respectively. $P_i^x$ and $P_i^w$ are the species' partial pressure in the carrier gas and at the liquid-gas interface, respectively. $J$ represents the total mass flux density.

### 2.4. Application of the Code to the Waste

The containment matrix composition, used in the code, is shown in Table 1. This matrix represents the major elements of the waste. The toxic elements (Pb or As) were introduced separately to this matrix. For example, in the case of As, the species selected in the code are:

**Table 1.** Mole numbers of elements present in the waste used in the computer calculation.

| Element | Na | Mg | Rh | Al | Ti | K | Ca | Ba | Ni | Si | Mn | Fe |
|---|---|---|---|---|---|---|---|---|---|---|---|---|
| Mole Numbers | 0.074 | 0.78 | 0.023 | 0.79 | 0.013 | 0.84 | 0.741 | 0.265 | 0.024 | 0.074 | 0.196 | 0.262 |

In the gas phase: $O_2$, $O$, $O_3$, $Al$, $AlC_2$, $Al_2C_2$, $AlO$, $AlO_2$, $Ba$, $BaO$, $Ba_2O$, $Ba_2O_2$, $C$, $C_2$, $C_3$, $CO$, $CO_2$, $C_2O$, $C_3O_2$, $Ca$, $CaO$, $Fe$, $Fe(CO)_5$, $FeO$, $K$, $K_2CO_3$, $KO$, $Mg$, $MgO$, $Mn$, $MnO$, $MnO_2$, $Na$, $Na_2$, $Na_2O_2$, $Ni$, $Ni(CO)_4$, $NiO$, $Rh$, $RhO$, $Si$, $SiC_2$, $Si_2C$, $SiO$, $SiO_2$, $Ti$, $TiO$, $As_4O_9$, $TiO_2$, $As_4O_6$, $AsO$, $AlAs$, $AsO_2$, $As$, $As_2O_3$, $As_2$, $As_4O_7$, $As_3$, $As_4O_8$, $As_4$, $As_4O_{10}$, Ar.

In the liquid phase: $AlAs$, $AlAsO_4$, $Ca_3(AsO_4)_2$, $As$, $As_2O_4$, $Ba_3(AsO_4)_2$, $FeAs$, $FeAsO_4$, $Mg_3(AsO_4)_2$, $MnAs$, $NiAs$, $Al$, $Al_2O_3$, $Ba$, $BaCO_3$, $BaO$, $C$, $Ca$, $CaCO_3$, $CaMgSi_2O_6$, $CaO$, $CaO*Al_2O_3*2SiO_2$, $CaO*MgO*2SiO_2$, $*2CaO*SiO_2$, $CaTiSiO_5$, $Fe$, $FeNaO_2$, $FeO$, $Fe_3O_4$, $*2FeO*SiO_2$, $FeO*TiO_2$, $FeSi$, $K$, $KAlO_2$, $K_2CO_3$, $K_2O$, $K_2O*SiO_2$, $Mg$, $MgO$, $MgO*Al_2O_3$, $MgSiO_3$, $Mg_2SiO_4$, $MgTiO_3$, $MgTi_2O_5$, $Mg_2TiO_4$, $Mn$, $MnO$, $Mn_2SiO_4$, $Na$, $Na_2CO_3$, $Na_2O$, $Na_2SiO_3$, $Ni$, $NiO$, $Rh$, $Si$, $SiC$, $SiO_2$, $Ti$, $TiC$, $Ti_2O_3$, $Ti_3O_5$, $Ti_4O_7$, $C$.

## 3. Results and Discussions

Figure 2 depicts the chemical composition of the waste before the treatment measured by XRF analysis. This figure shows that Ni, Fe, S, V, Ca, P, Sr and Mo are the waste's major constituents. The concentrations of elements present in the wastes were measured using ICP-AES, and were presented in Table 2. The elements with very low concentration are not listed in Table 2. The elements such as, Ni, V, Fe, Mg, Ca, Na, and Al represent the major constituents of wastes (100–1000 mg/kg), whereas the minor constituents (1–100 mg/kg) include Mo, Ba, Zn, Cr, Ti, Pb, and Mn. Toxic elements such as Cd and Sr, are present in the wastes but with relatively low concentrations. These results confirm the XRF analysis (Figure 2).

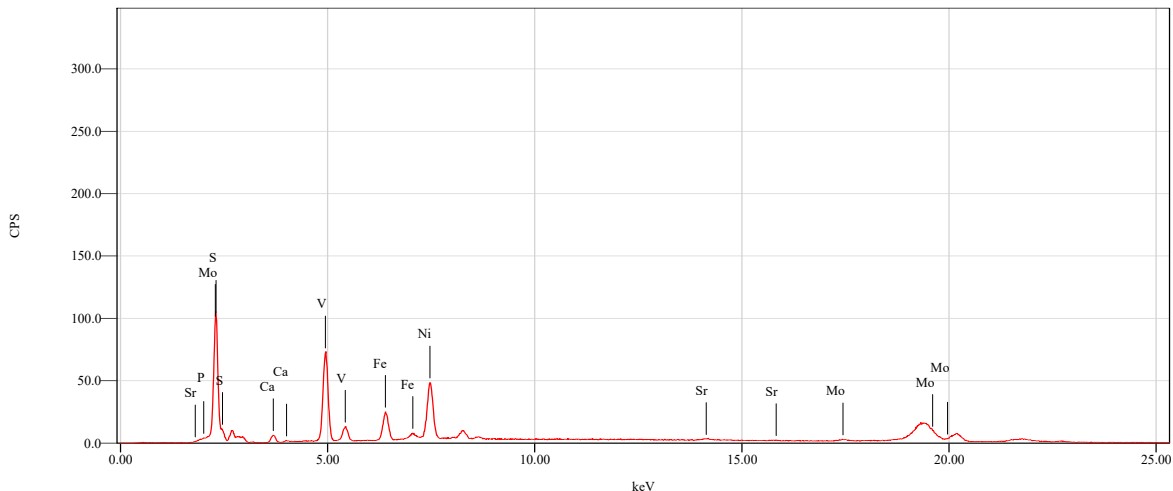

**Figure 2.** XRF analysis for Fly ash before treatment.

**Table 2.** Concentrations of elements present in the waste before treatment obtained by ICP.

| Element | V | Ni | Fe | Mg | Na | Ca | Al | Mo | Ba | Zn |
|---|---|---|---|---|---|---|---|---|---|---|
| C (mg/kg) | 5663 | 2420.2 | 2027 | 1917.4 | 587.4 | 386.5 | 135.5 | 51.5 | 33.4 | 22.8 |
| Element | Cr | Pb | Mn | Sr | Ti | Cu | Co | K | Ga | As |
| C (mg/kg) | 14.7 | 11.6 | 11.2 | 5.7 | 5.2 | 4.9 | 4.7 | 3.6 | 3.3 | 2.8 |

Figure 3a presents the EDX analysis of the waste before the treatment. The obtained results show that the waste consists mainly of carbon, but it also contains a significant amount of sulfur (4%). For this reason, the principal objective of this study is the separation of carbon present in the waste from metals. Several trials were carried out to optimize the best operating conditions for the waste treatment, such as plasma current and the flow rate of the air that was used as a carrier gas in the plasma. To obtain a maximum reduction of carbon, the fly ash collected in the filter bag was re-introduced in the furnace to complete the combustion of carbon. Initially, 1 kg of waste was introduced into the furnace, and the masses of the waste collected in the filter after the first, second and third cycle of treatment are, 696 g, 260 g, and 130 g, respectively. These results show that the waste's mass was reduced after each cycle, reaching a reduction of 85% during the third cycle, and this reduction is due to the combustion of carbon during the treatment.

After each cycle, the treated waste was analyzed with XRF, ICP, and EDX. Figure 3 predicts the principal elements remaining in the waste after each cycle of treatment. These Figures show that the waste kept these main constituents (NiO, $Fe_2O_3$, $SO_3$, $V_2O_5$, CaO, $P_2O_5$, SrO and $MoO_3$). EDX analysis, presented in Figure 3b–d, shows a reduction in the percentage of carbon in the waste, but the carbon remains the major constituent of the waste. Tables 3 and 4 give the concentrations of major and minor elements present in the waste after each cycle of treatment. The waste kept the same major and minor elements before and after treatment. The concentrations of major and minor elements were increased after each cycle of treatment. These results can be explained by the decreases of the concentration of carbon in the waste due to the combustion of carbon during the treatment.

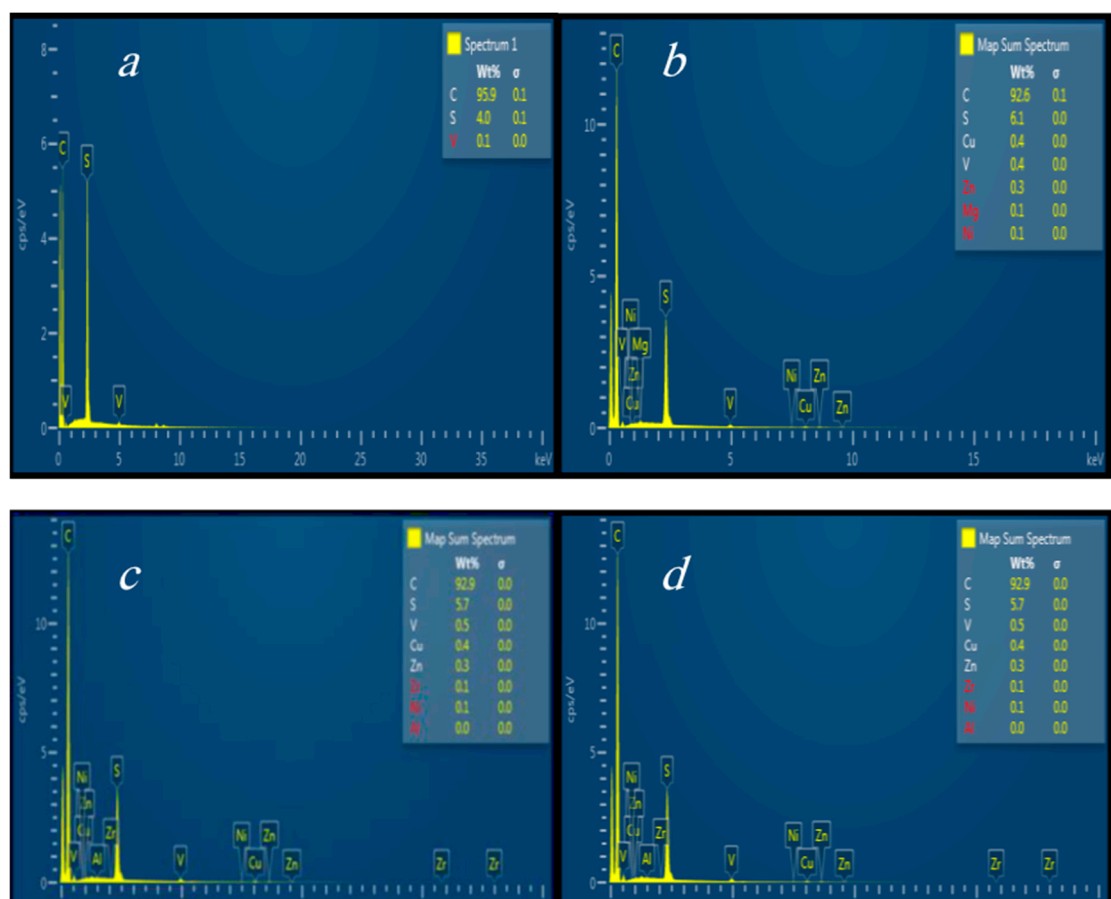

**Figure 3.** EDX analysis of fly ash (**a**) before treatment, (**b**) after 1st cycle of treatment, (**c**) after 2nd cycle of treatment, (**d**) after 3rd cycle of treatment.

**Table 3.** Concentrations of major elements present in the waste after treatment obtained by ICP.

| Element | | Na | Mg | Al | K | Ca | Ti | V | Cr | Fe | Ni | Cu | Zn | Mo |
|---|---|---|---|---|---|---|---|---|---|---|---|---|---|---|
| | 1st Cycle | 892.3 | 2449 | 585.6 | 109.4 | 878 | 72.1 | 5642 | 143.1 | 2280 | 2032 | 612.8 | 79.3 | 85.9 |
| C (mg/kg) | 2nd Cycle | 824.7 | 2505 | 696.1 | 125.2 | 381 | 65.8 | 6457 | 109.5 | 2569 | 2039 | 1006.2 | 93.8 | 79.6 |
| | 3rd Cycle | 1032. | 3064 | 427.6 | 104.7 | ND | 19.5 | 7358 | 123.6 | 2499 | 2561 | 1835.4 | 123.4 | 128 |

**Table 4.** Concentrations of minor elements present in the waste after treatment obtained by ICP.

| Element | | Mn | Co | Ba | La | Ga | Se | Sr | Zr | Ag | Cd | Sn | Ce | Pb | As |
|---|---|---|---|---|---|---|---|---|---|---|---|---|---|---|---|
| | 1st Cycle | 27.3 | 6.1 | 58.0 | 3.6 | 2.9 | 4.0 | 10.5 | 4.0 | 19.8 | 1.3 | 7.3 | 1.3 | 12.7 | 1.1 |
| C (mg/kg) | 2nd Cycle | 21.7 | 6.1 | 59.2 | 4.6 | 3.6 | 4.6 | 11.1 | 1.6 | 13.6 | 1.4 | 8.1 | 2.7 | 12.4 | 1.2 |
| | 3rd Cycle | 29.0 | 7.6 | 88.3 | 5.2 | 3.9 | 1.8 | 16.6 | 4.2 | 42.4 | 1.3 | 18.5 | 3.1 | 18.46 | 0.1 |

To understand the behavior of the elements, present in the waste, the masses of all elements after each cycle of treatment were calculated using the following equation:

$$m_k = C_k \cdot M_T \tag{5}$$

where $C_k$ and $m_k$ represent the concentration and the mass of element, respectively, and $M_T$ is the total mass of the waste used in each cycle.

Figure 4 shows that the masses of elements decrease after each cycle of treatment. This diminution can be attributed to the vaporization of these elements under the high

temperature of the plasma. These results were confirmed by the model calculation (Figure 5). This figure shows that the vaporization speed and the vaporized quantity are important for many elements, such as Ba, K, and Ti. Other elements like Mg, Rh, and Ca, are vaporized in the first 10 s, and after this time they are stable. This effect is due the combustion of carbon in the first 10 s. Elements like Fe, Ni, Si, and Al are not volatile and their mole numbers remains constant. These results are logical because the vaporization temperatures of these elements are very high. In contrast, the experimental results show a decrease in the quantities of Fe, Ni, and Al, after each cycle of treatment. This difference between the experimental measurements and the code results is due to the loss of waste in the components of the plasma system (furnace and exchanger) during the treatment, and because the initial used masses of these elements are very low (2.4 g for Ni, 2.01 g for Fe, and 0.135 g for Al).

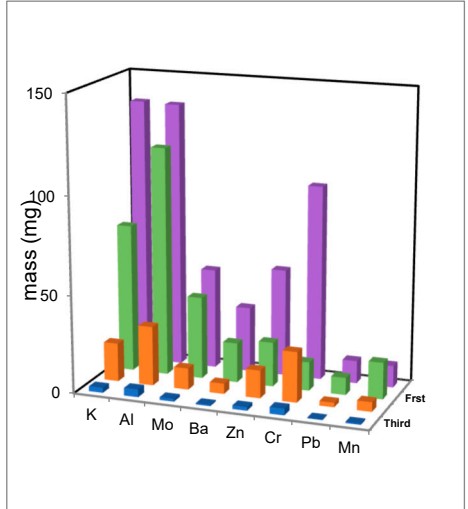 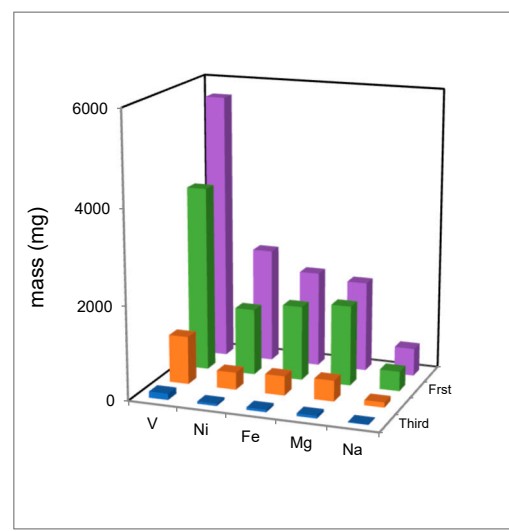

**Figure 4.** Mass (mg) of elements after each cycle of treatment.

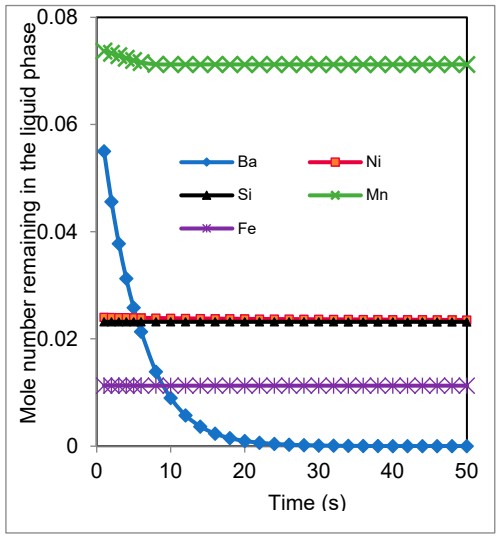 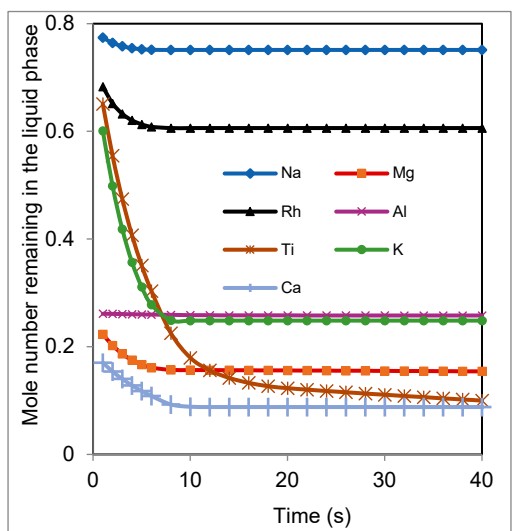

**Figure 5.** Variation of the mole number of elements present in the waste.

To know the compounds responsible on the volatility of the toxic elements, Figure 6 depicts the variation of titanium components in the liquid and gas phases. Initially, titanium was in the form of TiC (0.013 moles) and this compound disappears in the first second, which confirms that the combustion of carbon is responsible for the vaporization of Ti. Titanium continues to vaporize in the form of $TiO_2$ after the vanishing of carbon.

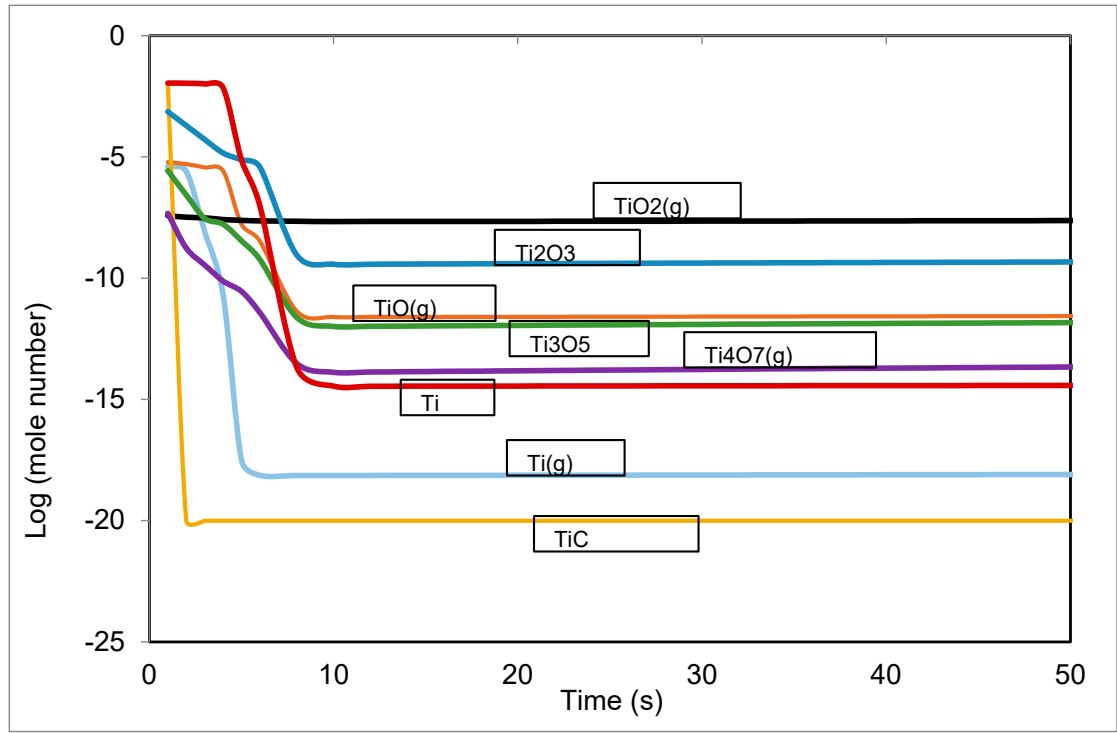

**Figure 6.** Variation of the mole number of titanium components in the liquid and gas phase.

Many parameters influenced the vaporization or the incorporation of toxic elements in the confinement matrix during the treatment. Of these parameters, we cited the bath surface temperature, the presence of oxygen in the carrier gas, and the composition of confinement matrix. Pb and As are chosen to study the effect of these parameters using our computer code. To have the same experimental conditions, the plasma current was fixed on the code at 250 A, and the total pressure P at $1.013 \times 10^5$ Pa.

The influence of the bath surface temperature on the arsenic volatility was presented in Figure 7. This figure shows that arsenic is not volatile for temperatures less than 2000 K, whereas for temperatures beyond 2100 K, the vaporization speed increases and the arsenic quantity that leaves the furnace becomes important. These results can be explained by the stability of the arsenic components with temperature. In fact, any increase in temperature in this range leads to the stability of arsenic gas species due to the decrease in their formation free enthalpies.

The liquid components of arsenic have the opposite behavior with temperature. The influence of temperature on lead volatility shows that any increase in temperature increases the vaporized quantity and the vaporization speed of lead.

The effect of the variation of oxygen pressure, on lead and arsenic volatility are presented in Figure 8. In this study, we used a temperature 2200 K for the experiments. These Figures show that the increase of the partial pressure of oxygen supported the incorporation of Pb and As in the containment matrix. In fact, when the atmosphere of the furnace becomes more oxidizing, the vaporized quantity and the speed of vaporization of As and Pb decrease.

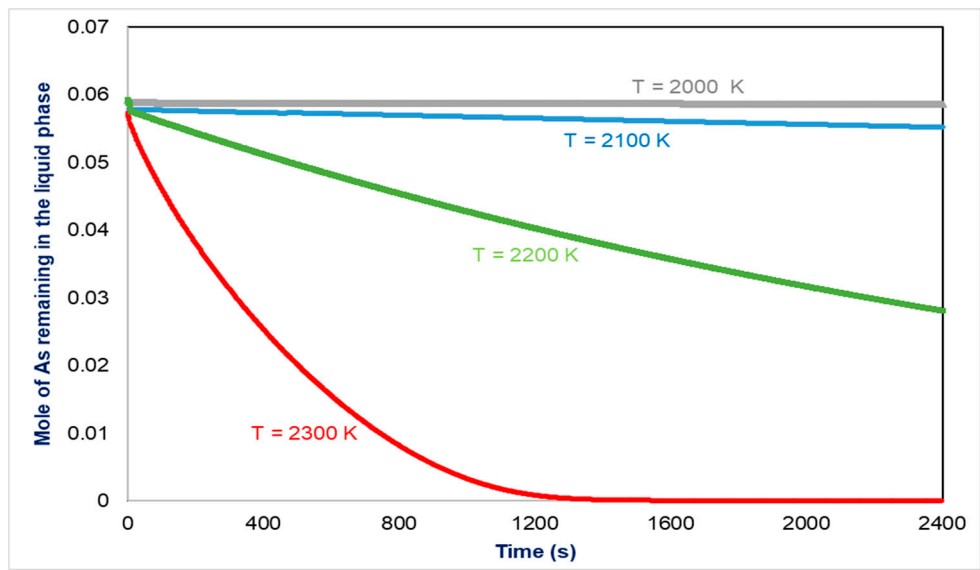

**Figure 7.** Influence of temperature on the volatility of As.

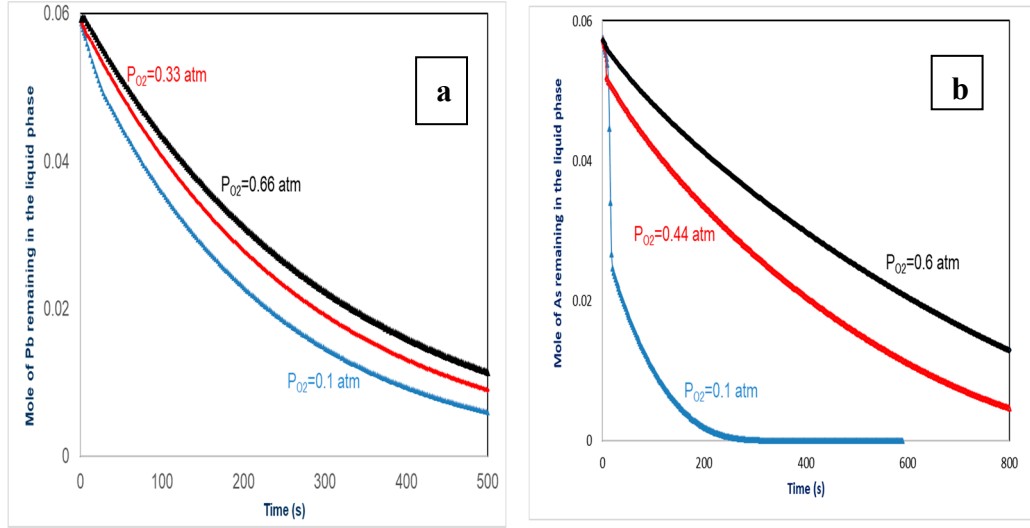

**Figure 8.** Influence of partial pressure of oxygen on the volatility of: (**a**) Pb, (**b**) As.

When the atmosphere is oxidizing, Equations (6) and (7) represent the principal reactions responsible for As volatility:

$$Ba_3(AsO_4)_2 = 2AsO \, (g) + 3Ba + 3O_2 \tag{6}$$

$$AlAsO_4 = AsO \, (g) + AlO_2(g) + 1/2O_2 \tag{7}$$

Equations (6) and (7) show that any addition of oxygen to the two reactions led to the formation of $Ba_3(AsO_4)_2$ and $AlAsO_4$. We conclude that the increase of the partial pressure of oxygen supported the confinement of As in the containment matrix.

Equations (8) and (9) represent the reactions of Pb vaporization when the atmosphere is oxidizing:

$$PbO = PbO \, (g) \tag{8}$$

$$PbO = Pb \, (g) + 1/2O_2 \tag{9}$$

Equation (9) shows that any addition of oxygen to the system moves the equilibrium to the formation of PbO and strengthens the incorporation of Pb in the containment matrix.

To study the influence of matrix composition on the volatility of As, three matrices were used. Matrix 3 represents the initial composition of waste and is given by Table 2. Matrix 1 and 2 have the same composition to matrix 1, but in matrix 1 the mole number of Ba was reduced to 0.01 mole, whereas in matrix 2 the mole number of Ba was increased to 0.5 mole. Figure 9 depicts the influence of the composition of the containment matrix on the arsenic volatility. The increase of the quantity of Ba in the containment matrix strengthens the confinement of As in the matrix. In fact, the addition of the mole number of Ba in the confinement matrix led to the increase of the mole numbers of $Ba_3(AsO_4)_2$ in the liquid phase. This molecule, which has a significant quantity (between $10^{-3}$ mol and $10^{-2}$ mol), is responsible for the confinement of As in the matrix.

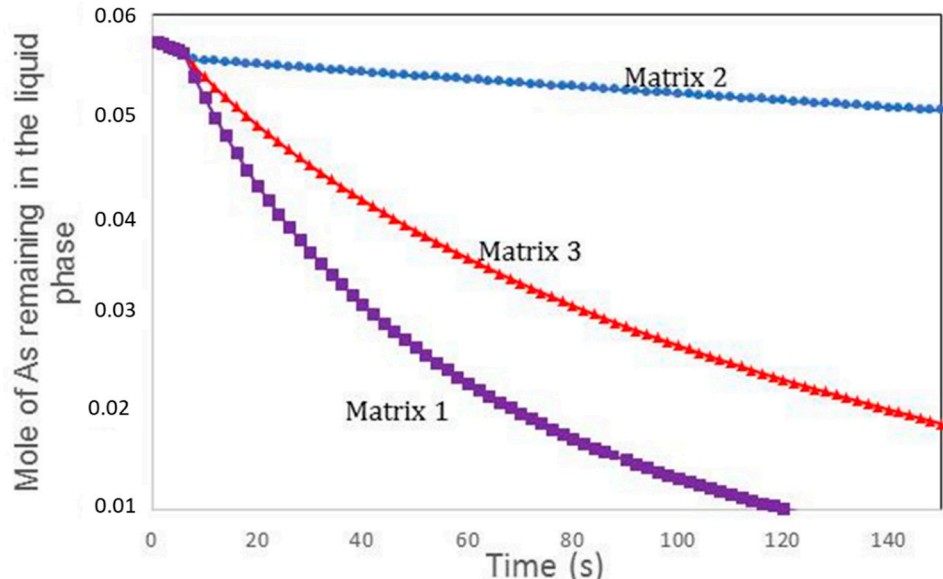

**Figure 9.** Influence of composition of confinement matrix on the volatility of As.

## 4. Conclusions

In this study, waste collected from power plants (that use oil as a source of fuel) was treated. EDX analysis shows that this waste is composed essentially of carbon and sulfur. This waste also contains metals, such as V, Ca, K, Mg, Al, Ba, Co, Ti, La, Se, Zr, Ag, Sn, Ce and Mn. ICP analysis confirmed the presence of toxic elements such as Cd, Sr, Pb, and As in this waste. The waste was treated using a plasma furnace to separate carbon from the metals. After three cycles of combustion, the mass of the waste was reduced by more than 85% and the metals were concentrated in the product collected in the filter bag. The volatility of toxic elements is due to the carbon combustion. The influence of parameters, such as the bath surface temperature, the presence of oxygen in the carrier gas, and the composition of the confinement matrix, on the volatility of Pb, and as was carried out using a computer code. These results show that arsenic remains in the liquid phase for temperatures less than 2000 K, whereas for temperatures beyond 2100 K, arsenic becomes very volatile. For lead, any increase of temperature increases its vaporized quantity and its vaporization speed. The addition of oxygen in the carrier gas leads to the heavy metal incorporation in the confinement matrix. The increase of the quantity of Ba in the containment matrix strengthens the confinement of as in the matrix.

**Author Contributions:** Conceptualization, I.A. and I.G.; methodology, I.A. and I.G.; software, I.G. and I.A.; validation, I.G.; formal analysis I.G.; investigation I.G.; data curation, I.A.; writing—original draft preparation, I.A. and I.G.; writing—review and editing, I.A., I.G., A.A. (Ahmed Alharbi) and A.A. (Abdullah Albeladi); visualization, A.A. (Ahmed Alharbi); supervision, I.A. and I.G.; project administration, I.A. All authors have read and agreed to the published version of the manuscript.

**Funding:** This research received no external funding.

**Institutional Review Board Statement:** Not applicable.

**Informed Consent Statement:** Not applicable.

**Data Availability Statement:** Data is contained within the article.

**Acknowledgments:** The authors acknowledge the financial support from energy and water research institute, King Abdullaziz City of science and Technology, project no.31-461, "Treatment of Waste from Desalination Plants Using Plasma".

**Conflicts of Interest:** The authors declare no conflict of interest.

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
