# Peer review of "Toxic Elements Behavior during Plasma Treatment for Waste Collected from Power Plants"

_applsci, doi:10.3390/app12136564_

Round 1

Reviewer 1 Report

 Dear Authors

I just recommend to add description about used XRF, EDX and ICP MS equipment.

Author Response

I just recommend to add description about used XRF, EDX and ICP MS equipment.

It is done.

Reviewer 2 Report

1. Authors need to state the main results/finding in the abstract.

2. English must be improved. Professional service is advised.

3. Can authors explain why actual experiment for the determination of volatility of toxic elements was not conducted, but instead using programming?

4. In Fig.1, what are the instruments denoted #1-5?

5. Names, specifications, and setup for equipment used for analysis must be included.

6. Can authors discuss why the main elements such as V, Ni, etc... were found in the wastes?

7. I can see that after 3 cycles of plasma treatment, the carbon was reduced by only 3% in comparison with non-treated wastes. Why is that? Is it sufficient?

8. Can authors explain why did some elements increase after the treatment, while other elements decrease? What is the factor?

9. In page 8, authors claimed that Ni decreased after treatment, however, I see that it actually increased (Table 3). Why?

10. Who did authors check your calculation in programming if they were correct? Any cross-examinations?

Author Response

  1. Authors need to state the main results/finding in the abstract.

It is done.

  1. English must be improved. Professional service is advised.

The English was revised.

  1. Can authors explain why actual experiment for the determination of volatility of toxic elements was not conducted, but instead using programming?

To control the volatility of the elements during plasma treatment of waste experimentally, we must use the emission spectroscopy method, but this method is complicated and difficult to apply in our case because the waste contains a large number of elements but we will try to use this technique in our next works

  1. In Fig.1, what are the instruments denoted #1-5?

It is indicated in Fig. 1.

  1. Names, specifications, and setup for equipment used for analysis must be included.

It is done.

  1. Can authors discuss why the main elements such as V, Ni, etc... were found in the wastes?

I don’t understand your question:

If you mean why these elements exist in the fly ash, because typical fuel oils contain Fe, Ni, V, and Zn.

If you mean why these elements don’t remain in the wastes after the treatment, I explained this point in the paper.

  1. I can see that after 3 cycles of plasma treatment, the carbon was reduced by only 3% in comparison with non-treated wastes. Why is that? Is it sufficient?

After the treatment the mass of waste was reduced to 85 % and the carbon represent more than 90 % of the weight of waste so we can conclude that the quantity of carbon evaporated exceed 80 % of the initial mass.

  1. Can authors explain why did some elements increase after the treatment, while other elements decrease? What is the factor?

I think you concluded this remark from tables 3 and 4. To better compare the quantity of matter after each cycle of treatment it is better to use the masses of the elements instead of their concentrations. According to figure 4 the masses of all the elements decreased after each cycle except in the case of Mn and Cr whose masses did not decrease in a regular way.

  1. In page 8, authors claimed that Ni decreased after treatment, however, I see that it actually increased (Table 3). Why?

I agree your remark but Ni decreases in mass (Fig. 4)  but not in concentration (g/Kg) (Table 3).

  1. Who did authors check your calculation in programming if they were correct? Any cross-examinations?

The code has been used for other cases such as the volatility of radioelements and heavy metals in previous works (see references) and the results obtained gave good correlations with the experimental results. In addition in this article the results are logical and in good agreement with the chemical laws

Reviewer 3 Report

The manuscript describes the problems of processing solid waste obtained from power plants that use oil as a fuel source. The waste was processed using thermal plasma technology. The aim was to separate the carbon in the waste from heavy metals and toxic elements. At the same time, the volatility of toxic gases was studied. A reduction in the weight of the waste of more than 85% after three cycles of plasma treatment was demonstrated. The effect of bath surface temperature, the presence of oxygen in the carrier gas and the composition of the confinement matrix on the volatility of Pb and As was also demonstrated.

The paper is valuable and well conceived as the topic of manuscript is actual and important. The paper provide logic and understandable form. The figures and tables are clear. I have the comments as follows:

·         Exhaust gas composition should be mentioned and discussed in the manuscript as well.

·         Figure 1 is not properly described in the text following the scheme. There are apparatus 1 to 5 identified but not described or identified.

·         Table 1, it would probably be more appropriate to provide the actual values of the concentrations of each particular element in the used waste instead of a molar number.

·         The cited references should be more updated. The last literature cited here is from 2013. There has certainly been a lot published on this topic in the last 10 years. There are journals such as Journal of Environmental Management, Journal of Cleaner Production, Journal of Environmental Chemical Engineering, Waste Management etc.

·         List of abbreviations would be helpful (even if most of abbreviations are introduced in the text directly)

Author Response

  • Exhaust gas composition should be mentioned and discussed in the manuscript as well.

      During the experiments we used a gas detection system which allowed us to know over time the nature and the quantity of the exhaust gases. We did not put these results because they are complicated and difficult to read.

  • Figure 1 is not properly described in the text following the scheme. There are apparatus 1 to 5 identified but not described or identified.

The title of Figure 1 is modified.

  • Table 1, it would probably be more appropriate to provide the actual values of the concentrations of each particular element in the used waste instead of a molar number.

I agree your remark but in the code I need the initial mole numbers of elements for this reason I used mole numbers in Table 1.

  • The cited references should be more updated. The last literature cited here is from 2013. There has certainly been a lot published on this topic in the last 10 years. There are journals such as Journal of Environmental Management, Journal of Cleaner Production, Journal of Environmental Chemical Engineering, Waste Management etc.

Recent references have been added in the introduction.

  • List of abbreviations would be helpful (even if most of abbreviations are introduced in the text directly)

I  agree your remark.

Round 2

Reviewer 2 Report

Authors have revised the manuscript and replied my comments/concerns with satisfactory. I believe now the manuscript is ready for publication.